# Safety of hyperbaric oxygen therapy in patients with heart failure: A retrospective cohort study

Simone Schiavo[1,2,3], Connor T. A. Brenna[1], Lisa Albertini[4], George Djaiani[1,2], Anton Marinov[2,5], Rita Katznelson[1,2,5]*

1 Department of Anesthesiology & Pain Medicine, University of Toronto, Toronto, ON, Canada, 2 Hyperbaric Medicine Unit, Toronto General Hospital, Toronto, ON, Canada, 3 Department of Anesthesia and Pain Management, University Health Network, Toronto, ON, Canada, 4 Department of Medicine, Division of Cardiology, University of Toronto, Toronto, ON, Canada, 5 Rouge Valley Hyperbaric Medical Center, Scarborough, ON, Canada

* rita.katznelson@uhn.ca

**Data Availability Statement:** The source data from our study cannot be shared publicly, in full, because of an ethical restriction levied by our institutional research ethics board. This is because

## Abstract

### Background

Hyperbaric oxygen therapy (HBOT) has several hemodynamic effects including increases in afterload (due to vasoconstriction) and decreases in cardiac output. This, along with rare reports of pulmonary edema during emergency treatment, has led providers to consider HBOT relatively contraindicated in patients with reduced left ventricular ejection fraction (LVEF). However, there is limited evidence regarding the safety of elective HBOT in patients with heart failure (HF), and no existing reports of complications among patients with HF and preserved LVEF. We aimed to retrospectively review patients with preexisting diagnoses of HF who underwent elective HBOT, to analyze HBOT-related acute HF complications.

### Methods

Research Ethics Board approvals were received to retrospectively review patient charts. Patients with a history of HF with either preserved ejection fraction (HFpEF), mid-range ejection fraction (HFmEF), or reduced ejection fraction (HFrEF) who underwent elective HBOT at two Hyperbaric Centers (Toronto General Hospital, Rouge Valley Hyperbaric Medical Centre) between June 2018 and December 2020 were reviewed.

### Results

Twenty-three patients with a history of HF underwent HBOT, completing an average of 39 (range 6–62) consecutive sessions at 2.0 atmospheres absolute (ATA) (n = 11) or at 2.4 ATA (n = 12); only two patients received fewer than 10 sessions. Thirteen patients had HFpEF (mean LVEF 55 ± 7%), and seven patients had HFrEF (mean LVEF 35 ± 8%) as well as concomitantly decreased right ventricle function (n = 5), moderate/severe tricuspid regurgitation (n = 3), or pulmonary hypertension (n = 5). The remaining three patients had

the data contains sensitive information from patient's medical charts (e.g., birth dates and personal health information). Taken together, this information may allow for the identification of individual study participants. We offer that an anonymized minimal data set can be prepared in aggregate and made available upon reasonable request via email to the study's first author (simone.schiavo@uhn.ca) or the Hyperbaric Medicine Unit, Toronto General Hospital, Toronto, Ontario, Canada (hyperbaricmedicineunit@uhn.ca).

**Funding:** The author(s) received no specific funding for this work.

**Competing interests:** I have read the journal's policy and the authors of this manuscript have the following competing interests: RK is a shareholder in the Rouge Valley Hyperbaric Medical Center, Toronto, ON. This does not alter our adherence to PLOS ONE policies on sharing data and materials.

HFmEF (mean LVEF 44 ± 4%). All but one patient was receiving fluid balance therapy either with loop diuretics or dialysis.

Twenty-one patients completed HBOT without complications. We observed symptoms consistent with HBOT-related HF exacerbation in two patients. One patient with HFrEF (LVEF 24%) developed dyspnea attributed to pulmonary edema after the fourth treatment, and later admitted to voluntarily holding his diuretics before the session. He was managed with increased oral diuretics as an outpatient, and ultimately completed a course of 33 HBOT sessions uneventfully. Another patient with HFpEF (LVEF 64%) developed dyspnea and desaturation after six sessions, requiring hospital admission. Acute coronary ischemia and pulmonary embolism were ruled out, and an elevated BNP and normal LVEF on echocardiogram confirmed a diagnosis of pulmonary edema in the context of HFpEF. Symptoms subsided after diuretic treatment and the patient was discharged home in stable condition, but elected not to resume HBOT.

## Conclusions

Patients with HF, including HFpEF, may develop HF symptoms during HBOT and warrant ongoing surveillance. However, these patients can receive HBOT safely after optimization of HF therapy and fluid restriction.

## Introduction

Hyperbaric oxygen therapy (HBOT) is an evidence-based intervention used to treat a variety of elective conditions, in addition to its role as an emergency treatment for carbon monoxide toxicity, decompression sickness, and arterial gas embolism (S1 Table) [1]. The safety profile of HBOT is very favorable: although minor side effects related to increased environmental pressure and/or systemic hyperoxia can occur (e.g., claustrophobia, transient myopia, or middle ear barotrauma) [2–4], serious treatment complications (e.g., seizures, pulmonary oxygen toxicity, or pulmonary edema) are extremely rare [5]. Anectodical evidence has suggested that patients with decreased left ventricular ejection fraction (LVEF) may be at an increased risk of acute heart failure (HF) during HBOT [6]. Although this risk has not been substantiated by robust evidence, left ventricular (LV) systolic dysfunction has traditionally been considered a relative contraindication to HBOT [6].

Several hemodynamic changes are known to occur during and immediately after hyperbaric oxygen exposure [7], and numerous mechanisms have been suggested to account for this [8,9]. The predominant effect is related to HBOT-induced vasoconstriction, the physiologically protective response to extremely high arterial partial pressures of oxygen [7], which increases systemic vascular resistance and thus cardiovascular afterload; this is associated with an increase in systolic and mean arterial blood pressure (BP). Cardiac output (CO) decreases due primarily to a decrease in heart rate (HR). Previous literature characterizing the effect of HBOT on CO is summarized in Table 1.

These hemodynamic changes appear to be well tolerated in patients without preexisting cardiac disease [15,16]. However, there is limited evidence regarding the applicability of HBOT in patients with HF and reduced ejection fraction (HFrEF) and, furthermore, no data on patients with HF with preserved ejection fraction (HFpEF) or HF with mid-range ejection

**Table 1. Previous studies characterizing the effect of hyperbaric oxygen therapy on cardiac output.**

| Study | CO change (% compared to baseline) | ATA |
|---|---|---|
| Whalen, 1965 [10] | -13 | 3.04 |
| Pisarello, 1987 [11] | -8 | 3.0 |
| | -15 | 2.5 |
| Pelaia, 1992 [12] | -17 | 2.2 |
| McMahon, 2002 [13] | -10 | 3.0 |
| Weaver, 2009 [14] | -18 | 2.5 |
| | -16 | 3.0 |

Changes in cardiac output associated with hyperbaric oxygen therapy among previous reports. Abbreviations: CO = cardiac output, ATA = absolute atmospheres of pressure.

fraction (HFmEF). We aimed to examine the safety of HBOT for patients with preexisting diagnoses of HF.

## Methods

### Study design

This is a retrospective cohort study of patients with HF who underwent elective HBOT between June 2018 and December 2020 in two Hyperbaric Medicine Centers in Ontario, Canada (Toronto General Hospital, Toronto; Rouge Valley Hyperbaric Medical Centre, Scarborough). Institutional Research Ethics Board approvals (CAPCR ID: 19–5081.1; IRB ID:2023-3194-14092-4) were obtained for study team members to collect data from medical records (last access to data on March 31, 2023; all authors but one (SS) were blinded to patient identities.

### Definitions

In accordance with the Canadian Cardiovascular Society guidelines [17], HF was defined as a clinical syndrome in which abnormal heart function results in (or increases the risk of) clinical symptoms and signs of reduced cardiac output and/or pulmonary or systemic congestion either at rest or with stress. Chronic HF represents the persistent and progressive nature of the disease, whereas acute HF is defined as a change in HF signs and symptoms resulting in the need for urgent therapy. Recent guidelines proposed a new and revised classification of HF according to LVEF [18–20], which includes: (i) HF with preserved ejection fraction (HFpEF) = LVEF $\geq$ 50%; (ii) HF with mid-range ejection fraction (HFmEF) = LVEF 41–49%; and (iii) HF with reduced ejection fraction (HFrEF) = LVEF $\leq$ 40%.

HFpEF is diagnosed in patients with signs and symptoms of HF as the result of high LV filling pressure, despite preserved LVEF ($\geq$ 50%) [18]. These patients also display normal LV volumes and an abnormal diastolic filling pattern (diastolic dysfunction) [18,21]; therefore, HFpEF is sometimes referred to as diastolic heart failure [22,23].

### Participants and data collection

We included all patients 18 years of age or older with a history of HF, regardless of EF, undergoing elective HBOT during the study period. History of HF was defined as a preceding diagnosis documented in and collected from the medical record, including cardiology assessments and reports. To further categorize these patients, LVEF measurements via echocardiography

were identified (where available) and used to stratify patients into three groups: (i) HFpEF = LVEF $\geq$ 50%; (ii) HFmEF = LVEF 41–49%; and (iii) HFrEF = LVEF $\leq$ 40% [17].

Each patient's demographic variables, past medical history, and medications were extracted from medical charts. Additional data extracted during the treatment period included HBOT indication, treatment pressure, total number of HBOT sessions, and adverse events associated with HBOT, including subjective symptoms reported by the patients and reported into the medical chart. All patients described in the study provided written consent to undergo HBOT for a clinical indication approved by Health Canada.

## Hyperbaric oxygen therapy protocol

Conventional HBOT protocols were utilized in the treatment of all patients, as previously described [15]: these included the administration of 100% oxygen at 2.0 or 2.4 atmospheres absolute (ATA) for 90 minutes, with 1–2 air breaks (0.21 fraction of inspired $O_2$ was supplied via a non-rebreather face mask at the same ATA for the treatments in mono-place chambers or by removing the plastic hood from the patient head during the treatments in the multi-place chamber) per session, five times weekly, either in a mono-place chambers (Sechrist 3600H and Sechrist 4100H, Sechrist Industries Inc., Anaheim, CA, USA; PAH-S1-3200, Pan-America Hyperbarics Inc., Plano, TX, USA; Sigma 36, Perry Baromedical, Riviera Beach, Fl, USA) or through a plastic hood in the multi-place chamber (rectangular Hyperbaric System, Fink Engineering PTY-LTD, Warana, Australia). Standard monitoring included measurements of systolic (SAP), diastolic (DAP), and mean (MAP) blood pressure (BP), heart rate (HR), and peripheral oxygen saturation (SpO2) assessed during a five-minute period preceding and following each HBOT session. BP was measured non-invasively using an upper arm cuff and automated sphygmomanometer (Connex VSM 6000, WelchAllyn—Hill-Rom, New York, NY, USA; Edan M3A Vital Signs Monitor, Edan Diagnostic, Inc. San Diego, Ca, USA) with the patient in a sitting or semi-sitting position.

## Outcomes

The objective of this study was to evaluate the safety of HBOT among patients with known HF. The primary outcome was to describe any clinical signs or symptoms of acute heart failure occurring during and immediately after HBOT. Secondary outcomes included other treatment complications, assessed as the number of patients experiencing HBOT-related adverse or serious adverse events, such as barotrauma, oxygen toxicity (either central nervous system or pulmonary), ocular changes, or confinement anxiety.

## Statistical analysis

Qualitative data including patient demographics and past medical history characteristics were summarized using descriptive statistics. Continuous data were expressed as means ± standard deviations.

## Results

### Clinical data

During the study period, 23 patients with a documented diagnosis of HF received elective HBOT. Table 2 summarizes patients' details and HBOT characteristics.

The mean patient age was 70 ± 12 years and 15 (65%) were male. A majority of patients had comorbid diagnoses of hypertension (21; 91%), type 2 diabetes (16; 70%), and/or coronary artery disease (14; 61%). At baseline, 13 (57%), 3 (13%), and 7 (30%) patients had HF

**Table 2. Baseline demographics, comorbidities, and medications of the patient cohort.**

| | | n = 23 |
|---|---|---|
| **Age (years)** | | **70 ± 12** |
| **Body Mass Index (kg/m$^2$)** | | **31 ± 11** |
| **Female** | | **8** |
| **Comorbidities** | | |
| History of hypertension | | 21 |
| Baseline Heart Failure classification: | | |
| *Preserved EF (LVEF ≥ 50%)* | | 13 |
| *Mid-range EF (LVEF 41–49%)* | | 3 |
| *Reduced EF (LVEF ≤ 40%)* | | 7 |
| Coronary artery disease | | 14 |
| Left ventricular hypertrophy | | 7 |
| Heart valvular disease | | 6 |
| Diastolic dysfunction | | 7 |
| Atrial fibrillation | | 9 |
| Peripheral vascular disease | | 11 |
| Diabetes mellitus: | | |
| *Type 1* | | 2 |
| *Type 2* | | 16 |
| Chronic obstructive pulmonary disease | | 5 |
| Restrictive lung disease | | 0 |
| Smoking status: | | |
| *Never* | | 15 |
| *Current* | | 2 |
| *Past* | | 6 |
| Renal insufficiency | | 14 |
| Dialysis | | 5 |
| **Medications** | | |
| ACEi/ARBs | | 11 |
| B-blockers | | 15 |
| Calcium channel blockers | | 13 |
| Diuretics | | 18 |
| Vasodilators | | 6 |
| | | 12 |
| **HBOT** Pressure (2.4 ATA) | | |

Descriptive analysis of patients included in this study (n = 23). Abbreviations: EF = ejection fraction, LVEF = left ventricular ejection fraction, ACEi = angiotensin-converting enzyme inhibitor, ARB = angiotensin receptor blocker, ATA = absolute atmospheres of pressure.

categorized as HFpEF (≥ 50%), HFmEF (41–49%) and HFrEF (≤ 40%), respectively. All 10 patients with HFrEF or HFmEF (100%) had a prior hospitalization for HF, compared to 7 out of 13 (54%) of patients with HFpEF. Overall, 11 (48%) were receiving treatment with ACEi/ARBs, 15 (65%) with betablockers, and 18 (78%) with diuretics, including 16 with loop diuretics, one with thiazide diuretics and one with potassium-sparing diuretics. Five (22%) patients were on dialysis, including one concurrently receiving diuretics, and only one patient with HFpEF was not receiving any diuretic nor dialysis. Pre-HBOT, all but one patient underwent a

**Table 3. Hyperbaric oxygen therapy details.**

| Patient # | LVEF (%) | ATA prescribed | Total number of treatments | Indication |
|---|---|---|---|---|
| 1 | 34 | 2.4 | 26 | AI LL |
| 2 | 54 | 2.4 | 60 | DFU |
| 3 | 66 | 2.4 | 35 | AI LL |
| 4 | 33 | 2.4 | 35 | STRI-RC |
| 5 | 55 | 2.4 | 23 | ORN (jaw) |
| 6 | 50 | 2.4 | 58 | DFU |
| 7 | 52 | 2.4 | 50 | DFU |
| 8 | 45 | 2.4 | 40 | AI LL |
| 9 | 31 | 2.0 | 49 | DFU |
| 10 | 50 | 2.4 | 60 | CPHYX |
| 11 | 24 | 2.4 | 33 * | CPHYX |
| 12 | 64 | 2.4 | 6 | DFU |
| 13 | 40 | 2.4 | 8 | CPHYX |
| 14 | 52 | 2.0 | 50 | DFU |
| 15 | 56 | 2.0 | 60 | DFU |
| 16 | 48 | 2.0 | 42 | DFU |
| 17 | 32 | 2.0 | 30 | DFU |
| 18 | 51 | 2.0 | 62 | DFU |
| 19 | 30 | 2.0 | 17 | DFU |
| 20 | 50 | 2.0 | 36 | DFU |
| 21 | | 2.0 | 60 | STRI-RP |
| 22 | 50 | 2.0 | 41 | CPHYX |
| 23 | 52 | 2.0 | 25 | DFU |

Treatment details for each patient included in the cohort (n = 23), including LVEF, HBOT exposure pressure, number of sessions, and indications for HBOT.
Abbreviations: HBOT = hyperbaric oxygen therapy; ATA = absolute atmospheres of pressure; LVEF = left ventricle ejection fraction; AI LL = arterial insufficiency–lower extremity; DFU = diabetic foot ulcer; STRI = soft tissue radiation injury; RC = radiation cystitis; RP = radiation proctitis; ORN = osteoradionecrosis; CPHYX = calciphylaxis.
*Patient #11: 7 out of the 33 sessions were at 2.0 ATA, and the remainder at 2.4 ATA.

transthoracic echocardiography (Table 2) in addition to a clinical assessment which excluded signs of acute heart failure prior to compression.

## HBOT characteristics

Twelve patients received HBOT at a pressure of 2.4 ATA; the remaining 11 patients underwent treatment at 2.0 ATA. Collectively, the 23 patients described in this study completed a total of 906 HBOT sessions. Each patient underwent an average of 39 ± 17 treatments, and half of them (434; 48%) were delivered at 2.4 ATA. Table 3 summarizes details of treatment for each patient.

## Acute cardiovascular complications

We observed symptoms consistent with HBOT-related HF in two patients (2/23, 9%). One patient with HFrEF (LVEF 24%) developed dyspnea after their fourth treatment for a diabetic foot ulcer. He had a history of hypertension, non-ischemic dilated cardiomyopathy, left ventricle hypertrophy, moderate pulmonary hypertension, mild tricuspid regurgitation, moderate diastolic disfunction, atrial fibrillation, peripheral vascular disease, obesity, diabetes, and kidney failure (not on dialysis). A routine random B-type Natriuretic peptide (BNP) collected one

month before HBOT was 402 g/mL (Lab reference range: < = 99.9 pg/mL). His hypertension was well controlled and a review of his BP measured before and after each session confirmed non-significant variations (mean pre-session SAP and DAP of 126 ± 12 mmHg and 72 ± 12 mmHg, respectively, and post-session SAP and DAP of 133 ± 15 mmHg and 75 ± 8 mmHg, respectively). Following the fourth HBOT, clinical examination revealed an increased work of breathing and crackles consistent with pulmonary edema, without peripheral oxygen desaturation. In the emergency department, his BNP was measured at 1580 pg/ml, and he was managed with increased oral diuretics but did not require hospitalization. This patient later disclosed that he had voluntarily held his diuretics before the treatment to avoid needing to urinate while inside the hyperbaric chamber. He subsequently continued HBOT, completing a total of 33 sessions without further complication.

A second patient, with HFpEF (LVEF 74%), developed dyspnea and desaturation after the sixth treatment session (also for a diabetic foot ulcer), ultimately requiring hospital admission. He had a history of hypertension, type 2 diabetes on insulin, obesity, coronary artery disease with HFpEF, and mild diastolic dysfunction (on double diuretic therapy). His hypertension was reported as well controlled on dual therapy (nifedipine and telmisartan), but a review of his BP measured before and after each session revealed a consistently increased SAP (mean 155 ± 11 mmHg) and normal DAP (77 ± 4 mmHg) before each treatment, and both an increased SAP (170 ± 3 mmHg) and DAP (86 ± 10 mmHg) following each treatment. During the acute episode following his sixth HBOT session, acute coronary ischemia and pulmonary embolism were clinically excluded. A diagnosis of pulmonary edema in the context of HFpEF was made on the basis of an elevated BNP (143 pg/mL), pulmonary congestion identified through bedside lung ultrasound and chest X-ray, and normal LVEF with the presence of diastolic dysfunction on transthoracic echocardiogram. The patient's symptoms subsided after administration of an intravenous loop diuretic (furosemide), and he was discharged home in stable condition. However, he elected not to resume HBOT. Three years later, he died of an unrelated oncologic pathology.

No acute cardiovascular complications were observed among the other 21 patients.

## Other complications

A total of seven non-serious adverse events were recorded: five instances of middle-ear barotrauma, and two of confinement anxiety. In each case, appropriate coaching and treatment were provided, and all patients continued HBOT without further complication.

## Discussion

In this study we investigated whether patients with a history of HF can safely receive HBOT. Two patients in our cohort (9%) experienced acute symptoms of heart failure in relation to HBOT. One had a history of HFrEF, which portends a theoretical risk with respect to HBOT. The other had a history of HFpEF, which has not been previously reported to increase cardiac risks of HBOT.

### Heart failure with reduced ejection fraction

HBOT is known to negatively impact cardiac output (CO), even among healthy patients [24]. The decreased CO, along with an increased afterload resulting from systemic vascular resistance, has been hypothesized to be the cause of pulmonary edema reported in patients with reduced EF [6]. In our cohort, among seven patients with HFrEF, only one developed signs of acute heart failure following HBOT. This patient had a severely impaired LVEF below 30% and he was receiving treatment with loop diuretics, although for two days he had been

withholding his morning doses. With appropriate coaching and therapy optimization (an increase in the dose of his loop diuretic), he continued HBOT and was able to complete 29 additional sessions without further complication. Our experience with these seven patients indicates that HBOT may exacerbate pulmonary congestion in patients with reduced ejection fraction, but also supports the feasibility of cautious treatment with close monitoring in this population after optimization of diuretic therapy. Interestingly, more recent studies have analyzed the long-term effects of HBOT on myocardial function, and paradoxically support a possible positive effect of HBOT on LVEF and other echocardiographic measures over longer time horizons [25–27].

## Heart failure with preserved ejection fraction

One of the 13 patients with HFpEF in our cohort developed acute signs of heart failure after six HBOT sessions. He had a history of hypertension, previous admission for heart failure, echocardiographic evidence of diastolic dysfunction, and ongoing treatment with thiazide diuretics but not loop diuretics. His consistently increased SAP and DAP after treatments may suggest a marked increase in afterload during and after each session [15], and increased afterload is a well-known effect of HBOT which contributes to decreases in CO [6,14]. Further, there is evidence that hyperoxia increases LV end-diastolic pressure (LVEDP), and it is associated with disturbances of both early and late phases of LV filling in patients with and without HF [28]. As a result, it is possible that a combination of increased afterload and impaired ventricular relaxation in the context of pre-existing diastolic disfunction might represent the mechanism of the pulmonary congestion exacerbation in this patient.

Complications of HFpEF resulting from HBOT have not been previously reported, although this finding is important as HFpEF is more prevalent among older adults, women, and those with obesity, systemic arterial hypertension, diabetes mellitus, and renal dysfunction [29]. Given the aging population and the increased medical complexity of patients seen in modern hyperbaric centres, the authors expect an increasing frequency of HBOT candidates with HFpEF in the hyperbaric medicine setting.

## Clinical implications

Our data suggest that a minority of patients with HF, regardless of EF, may develop acute heart failure symptoms. However, we also show that this event is rare and potentially preventable, and that these patients can complete HBOT safely after therapy optimization, with close surveillance before and after each session. Cardiac guidelines recommend the use of loop diuretics in patients with HFpEF and HFrEF, aiming to reduce symptoms of congestion [18,20]. In our study, no serious complications were observed among 21 of the 23 patients. Interestingly, all but one of these patients were either on therapy with loop diuretics or receiving regular dialysis. It is possible that optimizing medical therapy (e.g., initiating or titrating loop diuretics) for patients with HFpEF may avoid or further limit pulmonary congestion in the setting of HBOT. The same cardiac guidelines [18–20] also recommend strict BP control, as uncontrolled hypertension is a risk factor for acute exacerbation of HF, especially among patients with HFpEF. In our centers, we do not initiate HBOT if the baseline SBP > 180 or DBP > 100 mmHg. Furthermore, in circumstances where the BP is too labile after a treatment session, we defer the treatments until BP is evaluated and stabilized. This approach to BP management might have had an additional beneficial impact in decreasing the incidence of cardiac decompensation during HBOT in the cohort described herein.

Some patients with HF may present for HBOT without the typical history of HF symptoms and low LVEF, well known to physicians as pathognomonic of HFrEF. Indeed, HFpEF is

diagnostically challenging for a clinician, given the frequency of atypical symptoms and/or an unremarkable LVEF. In our study, 54% of patients with HFpEF did not have a prior hospitalization primarily caused by their HF, and the diagnosis was based on transthoracic echocardiogram and signs and/or symptoms of HF while undergoing investigations for other indications (e.g., acute coronary syndrome, or additional tests required during dialysis or diabetes management).

Therefore, even in the absence of a known impairment in LVEF, particular attention to any changes in the patient's clinical condition and pharmacological management during HBOT is warranted, and even mild-to-moderate respiratory or cardiac symptoms during HBOT should trigger further investigation to rule out an acute or subacute episode of HF. Patients with multiple comorbidities treated with numerous medications should be aware that any changes in their medications during HBOT should be discussed with their hyperbaric physician.

For the same reason, the availability of a baseline echocardiogram to facilitate evaluation of diastolic dysfunction during the initial assessment, rather than relying on other tests traditionally performed prior to HBOT (e.g., electrocardiogram or chest x-ray), may further reduce the risk of patients with unrecognized HF developing symptoms in the context of HBOT. However, there is currently a paucity of evidence to define the feasibility or cost-effectiveness of routine cardiac screening before HBOT to prevent these complications.

Finally, both patients who experienced HBOT-related HF in our study developed symptoms after several treatment sessions, rather than after the first one, suggesting the possibility of a cumulative effect of HBOT on pulmonary congestion (rather than acute onset, severe pulmonary edema in a patient who is incidentally referred for HBOT on the brink of this complication). This observation warrants particular consideration in the care of HF patients undergoing HBOT: despite undergoing several uneventful treatment sessions, these patients may gradually worsen, and still require close surveillance for the entire duration of treatment. Further research is needed to characterize the optimal management of patients with HF undergoing HBOT.

## Limitations

Our retrospective study has several inherent limitations. Because we retrospectively reviewed health records already compiled at the time of HBOT, it is possible that not all pertinent risk factors were identified and recorded. Our data relate to a cohort of patients treated in two urban centres, potentially limiting their generalizability to other settings; similarly, patients were treated by several different healthcare professionals at these settings, limiting consistency in measurement and reporting. Importantly, our study design cannot appreciate patients with HF who may have been referred for HBOT, assessed, and considered to be at too great a risk to proceed with treatment. Additionally, due to the rarity of patients with HF undergoing HBOT, we report on a small sample size, limiting estimates of the incidence of HF exacerbation related to HBOT, and subgroup analyses (e.g., stratified by EF %) present data on even smaller groups of patients. Finally, the primary outcome of the study was observational, and while two patients experienced symptoms of acute HF following HBOT with a close temporal relationship this cannot prove a causative relationship, especially considering the presence of possible confounding variables (e.g., types of and adherence to diuretics, changes in treatment pressure, and positioning after the complication).

## Conclusion

Patients with a history of heart failure, whether HFpEF or HFrEF, may develop symptoms of pulmonary congestion during or after HBOT. However, they can safely complete HBOT

following medical optimization with close attention paid to any clinical or pharmacological changes during treatment. Identifying patients at risk of HF exacerbation, and taking these measures to prevent acute symptoms during treatment, is an important objective of the pre-HBOT medical assessment.

## Supporting information

**S1 Table. Health Canada approved indications for hyperbaric oxygen therapy.** (DOCX)

**S1 Graphical abstract. Created with BioRender.com.** (JPG)

## Author Contributions

**Conceptualization:** Simone Schiavo, Rita Katznelson.

**Data curation:** Simone Schiavo, Connor T. A. Brenna, Anton Marinov, Rita Katznelson.

**Formal analysis:** Simone Schiavo.

**Methodology:** Simone Schiavo, Connor T. A. Brenna, Lisa Albertini.

**Project administration:** Simone Schiavo.

**Supervision:** Rita Katznelson.

**Validation:** George Djaiani.

**Visualization:** Simone Schiavo.

**Writing – original draft:** Simone Schiavo, Connor T. A. Brenna, Lisa Albertini.

**Writing – review & editing:** Simone Schiavo, Connor T. A. Brenna, Lisa Albertini, George Djaiani, Anton Marinov, Rita Katznelson.

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
