## [Editor Report · Decision Letter 0]

23 Oct 2023

PONE-D-23-31605Safety of hyperbaric oxygen therapy in patients with heart failure: a retrospective review.PLOS ONE

Dear Dr. Schiavo,

Thank you for submitting your manuscript to PLOS ONE. After careful consideration, we feel that it has merit but does not fully meet PLOS ONE’s publication criteria as it currently stands. Therefore, we invite you to submit a revised version of the manuscript that addresses the points raised during the review process.

We look forward to receiving your revised manuscript.

Kind regards,

Yashendra Sethi

Academic Editor

PLOS ONE

Additional Editor Comments:

Dear authors,

Thanks for submission - please address the following before we can proceed:

1. In your title, justify the use of retrospective review or kindly replace with retrospective longitudinal cohort study.

2.Please support your study with some digrammatic representation or summary as you may deem fit, it will allow readers a better understanding and attaract more viewership.

3. Revise methods and conclusions section of your abstract to be more clear.

---

## [Author Response · Author response to Decision Letter 0]

9 Nov 2023

Dear Academic Editor, please find the answers to all of your useful comments on the attached "Response to Reviewers - Revision Academic Editor2".

---

## [Decision Letter · Decision Letter 1]

1 Dec 2023

PONE-D-23-31605R1Safety of hyperbaric oxygen therapy in patients with heart failure: a retrospective longitudinal cohort studyPLOS ONE

Dear Dr. Schiavo,

Thank you for submitting your manuscript to PLOS ONE. After careful consideration, we feel that it has merit but does not fully meet PLOS ONE’s publication criteria as it currently stands. Therefore, we invite you to submit a revised version of the manuscript that addresses the points raised during the review process.

We look forward to receiving your revised manuscript.

Kind regards,

Yashendra Sethi

Academic Editor

PLOS ONE

Journal Requirements:

Additional Editor Comments:

Dear authors,

Thank you for your submission, we have now received comments from two subject experts and they have raised some minor concerns which need to be addressed before we can proceed further.

Reviewers' comments:

Reviewer's Responses to Questions

**Comments to the Author**

1. If the authors have adequately addressed your comments raised in a previous round of review and you feel that this manuscript is now acceptable for publication, you may indicate that here to bypass the “Comments to the Author” section, enter your conflict of interest statement in the “Confidential to Editor” section, and submit your "Accept" recommendation.

Reviewer #1: (No Response)

Reviewer #2: All comments have been addressed

2. Is the manuscript technically sound, and do the data support the conclusions?

Reviewer #1: Partly

Reviewer #2: Yes

3. Has the statistical analysis been performed appropriately and rigorously? 

Reviewer #1: N/A

Reviewer #2: Yes

4. Have the authors made all data underlying the findings in their manuscript fully available?

Reviewer #1: No

Reviewer #2: Yes

5. Is the manuscript presented in an intelligible fashion and written in standard English?

Reviewer #1: Yes

Reviewer #2: Yes

6. Review Comments to the Author

Reviewer #1: The study delves into a crucial issue that could significantly impact our capacity to administer HBOT to deserving patients who might otherwise be excluded. While it is well designed and well written, there are a few issues that should be addressed before its publication:

Line 64: “HBOT increases cardiovascular afterload, with associated increases in systolic and mean arterial blood pressure (BP), while cardiac output (CO) decreases due primarily to a decrease in heart rate (HR).” And line 73: “Numerous mechanisms for the effect of HBOT on CO have been suggested although this effect predominately results from HBOT-induced vasoconstriction, the physiological protective response to extremely high arterial partial pressures of oxygen”

Please consider merging these sentences for more coherent explanation regarding HBOT’s effect on CO:

high oxygen level (pressure?) � vasoconstriction � increases cardiovascular afterload…

Line 85: Please consider omitting “longitudinal”

Retrospective cohort studies are always 'longitudinal,' because they examine health outcomes over a span of time.

Line 109: “We included all patients 18 years of age or older with a history of HF”.

As listed in their medical charts? based on baseline echocardiography? Based on their symptoms history?

Please be more specific.

Line 125: How did you provide air breaks in a mono-place chamber? Please describe the technique. In a multi-place chamber, air breaks are accomplished simply by removing the mask, but in a mono-place chamber, a complete exchange of the chamber’s gas may be required.

Line 129: Standard monitoring included measurements of systolic (SAP), diastolic (DAP), and mean (MAP) blood pressure (BP), heart rate (HR), and peripheral oxygen saturation (SpO2) assessed during a five-minute period preceding and following each HBOT session.

Consider providing data regarding the blood pressure of the study cohort, including baseline values and the mean change from baseline following HBOT. Were there any changes in BP during the index HBOT session for subjects who developed HF exacerbation?

Clinical implications:

Describe your practice following BP evaluation before treatment. Do you exclude patients with uncontrolled hypertension from entering the session? Such a practice may impact CHF exacerbation rates and should be recommended if applicable.

Table S1:

Please add title to the table.

Are the mentioned indications approved by Health Canada? By FDA?

Reviewer #2: Article is an excellent article.worth publishing.Any role of prophylactic diuretics and NT PRO BNP measurements before starting the HBOT??Superb article.Congratulations.

7. PLOS authors have the option to publish the peer review history of their article (what does this mean?). If published, this will include your full peer review and any attached files.

Reviewer #1: **Yes: **Dr. Keren Doenyas-Barak

Reviewer #2: **Yes: **Dr Siddharth Gosavi

---

## [Author Response · Author response to Decision Letter 1]

21 Dec 2023

Dear reviewers, please find attached the responses to your valuable comments, on the attached "Revision reviewers_HBO in HF 2023 - final".

---

## [Editor Report · Decision Letter 2]

27 Dec 2023

Safety of hyperbaric oxygen therapy in patients with heart failure: a retrospective cohort study

PONE-D-23-31605R2

Dear Dr. Schiavo,

We’re pleased to inform you that your manuscript has been judged scientifically suitable for publication and will be formally accepted for publication once it meets all outstanding technical requirements.

Kind regards,

Yashendra Sethi

Academic Editor

PLOS ONE

Additional Editor Comments (optional):

Thank you for addressing all the concerns. We can now accept the revised version for publication.

Wishing you a merry Christmas and a happy new year!
---

## [Editor Report · Acceptance letter]

31 Jan 2024

PONE-D-23-31605R2 

PLOS ONE

Dear Dr. Katznelson, 

I'm pleased to inform you that your manuscript has been deemed suitable for publication in PLOS ONE. Congratulations! Your manuscript is now being handed over to our production team.

Kind regards, 

on behalf of

Dr. Yashendra Sethi 

Academic Editor

PLOS ONE